# Wild primates copy higher-ranked individuals in a social transmission experiment

Charlotte Canteloup[1,2,3]*, William Hoppitt [4,5,6] & Erica van de Waal [1,2,3,6]

Little is known about how multiple social learning strategies interact and how organisms integrate both individual and social information. Here we combine, in a wild primate, an open diffusion experiment with a modeling approach: Network-Based Diffusion Analysis using a dynamic observation network. The vervet monkeys we study were not provided with a trained model; instead they had access to eight foraging boxes that could be opened in either of two ways. We report that individuals socially learn the techniques they observe in others. After having learnt one option, individuals are 31x more likely to subsequently asocially learn the other option than individuals naïve to both options. We discover evidence of a rank transmission bias favoring learning from higher-ranked individuals, with no evidence for age, sex or kin bias. This fine-grained analysis highlights a rank transmission bias in a field experiment mimicking the diffusion of a behavioral innovation.

[1] Department of Ecology and Evolution, University of Lausanne, 1015 Lausanne, Switzerland. [2] Inkawu Vervet Project, Mawana Game Reserve, KwaZulu Natal 3115, South Africa. [3] Anthropological Institute and Museum, University of Zurich, Winterthurerstrasse 190, 8057 Zurich, Switzerland. [4] Faculty of Biological Sciences, University of Leeds, Leeds, UK. [5] School of Biological Sciences, Royal Holloway, University of London, London, UK. [6]These authors contributed equally: William Hoppitt, Erica van de Waal. *email: charlotte.canteloup@gmail.com

Social learning can be broadly defined as "learning that is facilitated by observation of, or interaction with another individual or its products"[1,2], and can result in one individual learning a behavior pattern after being exposed to another individual performing it. Such social transmission can result in the spread or "diffusion" of novel behavior through groups, and, potentially, the formation of group-specific traditions and culture[3]. This has led to interest in the extent to which social transmission occurs in wild populations, what determines the pathways of diffusion and the factors that determine when stable traditions and cultures arise[2,4].

The study of social learning has been recently enriched by the integration of social networks analysis and associated statistical methodologies[5]. Such complementary approaches allow researchers to test for social transmission in wild freely interacting populations of animals and track the diffusion of behavioral innovations across social networks such as the spread of "lobtail feeding" in humpback whales[6], new tool use in chimpanzees[7] and bottlenose dolphins[8] or new foraging techniques in great tits[9]. Social network structure has been found to shape the diffusion of socially learned behaviors; association of individuals in close proximity can be used to predict innovation spread[10]. Socially central individuals (i.e., those who have themselves numerous and strong network ties) have been reported to be the more likely to witness a task and socially acquire information[11,12].

The maintenance of cultural behaviors has been proposed to be partly due to an efficient and accurate propagation of new knowledge[13,14]. Some have suggested that particular mechanisms of social learning, such as imitation, are required for culture, since they allow high-fidelity copying of behavior[15], e.g., when a foraging technique is observed, the observer socially learns the same foraging technique and not a different one with the same function. Alternatively, culture may be primarily reliant on processes that insulate behavior from change between acquisition and re-transmission, e.g., once an observer learns a specific foraging technique, it continues to use that technique rather than asocially learning an alternative technique[16].

Others have suggested that culture may be reliant on the social-learning strategies employed, i.e., learning biases based on the content of behavior or its context[4]. Content biased social-learning strategies focus on characteristics of the observed behavior (e.g., bias towards better payoff). Context biases focus on other cues, such as their frequency in a population (e.g., copy rare behaviors; copy the most common behavior or the one exhibited by the majority of individuals: conformity bias) or particular traits of models, which likely correlate with fitness (termed "model biases" or "who strategies" e.g., copy kin, dominant, aged individuals)[17]. Note that we use "copying" as a synonym for "social learning"[4] that results in matching behavior between demonstrator and observer regardless of the underlying implied psychological mechanisms.

Several primate species have been shown to be selective in copying specific individuals or behaviors, revealing surprising parallels with the social learning of humans[18–20]. Captive chimpanzees displayed bias towards copying older, high-ranking individuals[21] and the majority of individuals[22]. Field experiments reported multiple social-learning strategies in wild vervet monkeys in which infants processed food in the same way as their mother[23,24], females—the philopatric sex—were preferred as models than males[25] and dispersing males conformed to the local foraging norm, abandoning their native preference[26]. By contrast, wild vervet monkeys showed no preference to copy a high-ranking adult female compared to a lower-ranking one[27]. Capuchins have been found to copy older individuals with more successful fruit opening techniques[28] while in vervet monkeys, males, but not females, were more likely to copy the male model receiving a higher payoff in a two-action learning task[29].

An overall copying bias might arise as a result of any combination of three component biases: performance bias, attention bias and social information use bias. An individual might be copied more than other potential demonstrators because it performs the behavior more (performance bias), and/or because its actions are more likely to be observed than the actions of others (attention bias). For example, Older and higher-ranking capuchins have been found to be more frequently watched by conspecifics while cracking nuts[30]; vervet monkeys have been reported to show a greater attention to female models[25] but not to dominants[27,31], and juveniles have been reported to pay more attention to their maternal relatives when foraging[31]. The performance and attention bias effects multiplied together determine the rate at which an individual is observed. Alternatively, it could be that each observation of behavior has a different weight on the observer's behavior, dependent on some characteristic of the observed individual, e.g., observations of higher-ranked individuals may be more likely to be copied than those of lower-ranked individuals (social information use bias). In this paper, we focus primarily on isolating the social information use bias, by using a statistical model that allows us to estimate the social transmission effect, per observation, between different classes of individuals.

Despite promising indications of primate social-learning strategies, some inconsistencies have been reported[32,33] and the interpretations of some findings as indicative of social-learning strategies have been controversial[34]. Moreover, many previous studies tested subjects by imposing a trained model who uses one technique in a two-alternative-actions design (henceforth termed "options"). Such a design is powerful in identifying social learning but may limit our understanding of who first solves a task in a group and whether these solvers are preferred as models, as well as if and how techniques spread across groups in realistic contexts. The majority of these earlier studies looked for evidence of a single social-learning strategy and did not evaluate whether other and multiple potential alternative explanations were plausible. Therefore, more open diffusion experiments, which allow the animals to freely interact during the course of the diffusion, are required to test whether these strategies still operate when superimposed on primates' social networks.

The present field study investigates the social transmission of novel foraging behavior in a nonhuman primate species: vervet monkeys (*Chlorocebus pygerythrus*). For this, we offer to the monkeys simultaneously eight "two-option" puzzle boxes, named "artificial fruits", that could be opened in either of two different ways to gain access to a small apple slice: a drawer in the front could be pulled and a door on the top of the box could be lifted (Fig. 1). We test two groups of monkeys that were free to interact with the boxes and to discover by themselves how to open the boxes using one of the two opening techniques. We record the exact time of each manipulation and the identity of observers allowing us to have fine-grained observation data to construct a dynamic observation network, and to quantify the "per observation" social effect between different classes of individual. While association networks[6,35–37], genetic networks[8], and static observation networks[11] have been used so far in modeling social transmission, very few studies used a dynamic observation network[7,28]. We aim here to challenge the notion of a single best strategy or a strategy associating several biases while previous studies on vervet monkeys reported a "copy adult females" strategy[23–25]. First, we report that vervet monkeys socially learn a novel foraging technique by observing their groupmates. Second, according to the "directed social learning" hypothesis[38], higher-ranked individuals are more influential demonstrators than lower-ranked individuals. Third, learning one option promotes asocial learning of the other option, making monkeys 31x more likely to asocially learn the second option after having socially learnt the

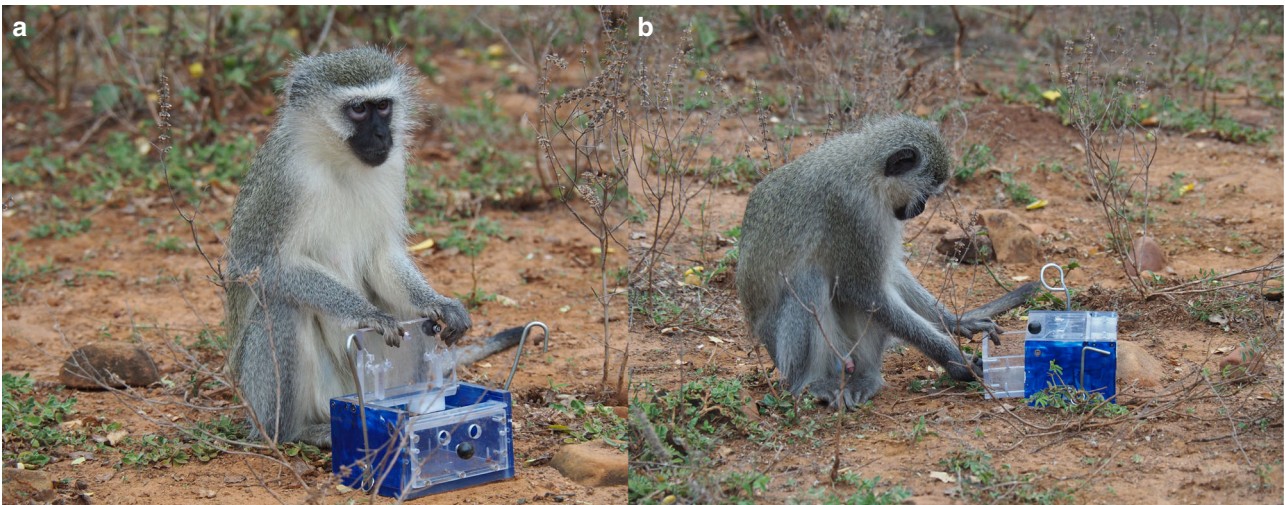

**Fig. 1 Experimental design. a** Adult female opening the box with the "lift" technique. **b** Juvenile male opening the box using the "pull" technique. Photographs copyright: Charlotte Canteloup.

first one than individuals naive to both options. By combining ecologically valid field experiments and a powerful modeling approach, we aim to shed new light on the social-learning strategies that may drive cultural transmission in a wild primate.

## Results

**Wild vervet monkeys socially learn a novel foraging technique.** We ran open diffusion experiments in two groups of vervet monkeys: Noha ($N = 28$) and Kubu ($N = 12$). In Noha, all group members touched one box at least one time with a latency of first contact ranging from 129 s to 89,954 s (mean = 19,856.04 s ± standard error of the mean = 4500.59 s; Supplementary Table 1 in Supplementary Note 1). In Kubu, all group members touched one box at least one time with a latency of first contact ranging from 184 s to 49,546 s (mean = 13,333.73 s ± standard error of the mean = 4248.29 s; Supplementary Table 1 in Supplementary Note 1). In Noha, the first individual who succeeded to open one box was the highest-ranked adult female, using the lift technique ("Gene"; Table 1 and Fig. 2a) and, in Kubu, it was the highest-ranked adult male using the pull technique ("Lif"; Table 1 and Fig. 2b). In Noha, 19 individuals of 28 succeeded to open the box using one option at least one time and, in Kubu, 10 of 12 did so (Table 1 and Fig. 2a, b).

We used network-based diffusion analysis (NBDA)[36,39,40], a specialized statistical method for detecting and quantifying social transmission and identifying the typical pathways of social transmission, which can also be expanded to investigate social-learning strategies[41] (see Methods section and "Model description" in Supplementary Note 2 for further details). We used the order of acquisition diffusion analysis (OADA) variant of NBDA that takes as data the order in which individuals in each group acquired a new behavioral trait. The underlying assumption is that the rate of social transmission between individuals, if it occurs, is proportional to the network connection between them; the more two individuals are connected (i.e., observe each other), the more opportunities they have to learn from each other. A parameter, $s$, is fitted to the data estimating the rate of transmission per unit of network connection, with $s = 0$ representing the null hypothesis of only asocial learning. Multiple networks, representing different pathways, can be included in the model allowing researchers to test for differences in rates of transmission on different pathways[41].

We extracted a dynamic observation network from open diffusion experiment data reflecting the number of times each individual observed others performing the tasks prior to that point in time, directly quantifying opportunities for social learning. Contrary to static networks, dynamic ones have the advantage of capturing temporal aspects of acquisition[42], thus the fact that not all individuals can learn a task at a specific point in time. Following the example given by Hoppitt "Imagine a group of three individuals: A, B and C. A learns the behavior first. Next, B observes A performing the behavior three times and then learns the behavior. Finally, C observes A performing the behavior four times and subsequently learns the behavior last. A static network would represent the network as having links of strength 3 from A to B and 4 to C, so an NBDA model based on this network would predict that C was more likely to learn second. In reality, we might expect B to be more likely to learn second, because B observed A performing the behavior first"[41], using dynamic networks can help reveal social transmission that may be obscured in a summarized static network. Dynamic networks have been reported to be more powerful than static ones[7] and have been rarely used in the literature[7,28], making our study particularly relevant. We built models testing whether the dynamic observation network predicted the order with which individuals acquired both opening techniques. In this sense, the model predicts the likelihood of learning each option for each individual, updating it after each learning event. In these models, we included networks representing two different pathways of learning: option-specific social learning and cross-option social learning. Option-specific (OS) social learning (quantified by $s_{os}$) occurs when observation of a task solution using the lift option increases the rate at which the observer learns the lift option, and likewise for the pull option. Conversely, cross-option (CO) social learning (quantified by $s_{co}$) occurs when observation of a task solution using the lift option increases the rate at which the observer learns the pull option, and vice versa. Therefore, a model in which $s_{os} = s_{co}$ represents the hypothesis that social transmission generalizes completely between the two options, e.g., by large scale local enhancement attracting observers to the task. $s_{os} > s_{co}$ represents the hypothesis that social transmission is specific to the option observed (to the extent quantified by $s_{os} - s_{co}$) for example due to social learning of a specific technique superimposed on large scale local enhancement. $s_{os} > 0$, $s_{co} = 0$ represents the case where social transmission is completely option specific with no generalization to the other option (high-fidelity copying) (see Supplementary Note 2 for more information). Thus, use of these two networks enables us to establish the level of copying fidelity in the task.

**Table 1 Composition of the two study groups Noha (NH) and Kubu (KB).**

| Group | Individual | Age | Sex | Rank | Nb observers | Order of acquisition first success (L or P) | Order of acquisition lift option | Order of acquisition pull option |
|---|---|---|---|---|---|---|---|---|
| Kubu | Aare | Adult | Female | 6 | 1 | 5 (L) | 7 | NA |
| Kubu | Amur* | Adult | Female | 7 | 0 | 9 (L) | 9 | NA |
| Kubu | Arn | Sub-adult | Male | 9 | 8 | 8 (L) | 4 | 5 |
| Kubu | Avo | Sub-adult | Male | 5 | 9 | 2 (L) | 1 | 3 |
| Kubu | Lif | Adult | Male | 1 | 9 | 1 (P) | 5 | 1 |
| Kubu | Mal | sub-adult | Male | 8 | 5 | 7 (P) | 3 | 2 |
| Kubu | Mis | Sub-adult | Male | 10 | NA | NA | NA | NA |
| Kubu | Nessi | Adult | Female | 11 | 6 | 4 (L) | 6 | 6 |
| Kubu | Yalu | Adult | Female | 4 | 10 | 3 (L) | 2 | 4 |
| Kubu | Yan | Sub-adult | Male | 3 | 5 | 6 (L) | 8 | NA |
| Kubu | Yeni* | Adult | Female | 2 | 2 | 10 (L) | 10 | NA |
| Noha | Bela* | Sub-adult | Female | 24 | 1 | 17 (L) | 16 | NA |
| Noha | Bos | Sub-adult | Male | 11 | NA | NA | NA | NA |
| Noha | Can | Adult | Male | 3 | NA | NA | NA | NA |
| Noha | Gaya | Adult | Female | 2 | 9 | 11 (L) | 11 | 10 |
| Noha | Gene | Adult | Female | 1 | 25 | 1 (L) | 1 | 1 |
| Noha | Gla | Sub-adult | Male | 4 | 24 | 9 (L) | 8 | NA |
| Noha | Gran | Sub-adult | Female | 10 | 25 | 8 (P) | 9 | 6 |
| Noha | Jixi | Sub-adult | Male | 20 | 26 | 4 (P) | 4 | 2 |
| Noha | Lima | Sub-adult | Female | 22 | 3 | 13 (L) | 13 | 11 |
| Noha | Prai | Sub-adult | Female | 17 | 1 | NA | NA | NA |
| Noha | Pret* | Adult | Female | 13 | 4 | 15 (L) | 19 | NA |
| Noha | Pro | Sub-adult | Male | 19 | NA | NA | NA | NA |
| Noha | Pru* | Sub-adult | Male | 16 | 8 | 14 (L) | 14 | NA |
| Noha | Renn* | Sub-adult | Female | 25 | 5 | 18 (L) | 17 | 12 |
| Noha | Reva | Adult | Female | 27 | 22 | 6 (P) | 6 | 4 |
| Noha | Rey | Sub-adult | Male | 28 | NA | NA | NA | NA |
| Noha | Rhe | Sub-adult | Male | 7 | 25 | 5 (P) | 5 | 3 |
| Noha | Roma | Adult | Female | 23 | NA | NA | NA | NA |
| Noha | Rosl | Sub-adult | Female | 26 | NA | NA | NA | NA |
| Noha | Tir* | Sub-adult | Male | 14 | 3 | 16 (L) | 15 | NA |
| Noha | Twe | Adult | Male | 6 | NA | NA | NA | NA |
| Noha | Uji | Sub-adult | Male | 12 | 8 | 12 (L) | 12 | NA |
| Noha | Ula* | Sub-adult | Male | 15 | 5 | 19 (L) | 18 | NA |
| Noha | Umt | Sub-adult | Male | 21 | 15 | 7 (L) | 7 | 9 |
| Noha | Upps | Adult | Female | 5 | 14 | 2 (L) | 2 | 5 |
| Noha | Xala | Sub-adult | Male | 9 | 8 | 3 (L) | 3 | 8 |
| Noha | Xian | Adult | Female | 8 | 24 | 10 (L) | 10 | 7 |
| Noha | Zan | Sub-adult | Male | 18 | NA | NA | NA | NA |

Individual level variables (group; individual, age, sex, rank), the number of observers (obs.) over the course of the experiment, the order of acquisition of the first success (lift or pull), the order of acquisition of the lift technique; the order of acquisition of the pull technique. NA indicates that the individual did not succeed in opening the box. Individuals marked with "*" are the ones who were opportunistically tested at the end of the experiment when the successful solvers were not around.

There was strongest support for exclusively OS social transmission (63.8% of chance that this model is the best one) followed by models in which OS social transmission was stronger than CO social transmission (22.2%; see Supplementary Fig. 1 in Supplementary Note 2). There was little support for models in which social transmission generalized between the two options (9.0%), nor for models in which there was only CO social transmission (2.5%; see Supplementary Fig. 1 in Supplementary Note 2). We also found some weak evidence that females were faster at social learning than males (see Table 2, and Supplementary Table 1 in Supplementary Note 1), which can be linked to the fact that females are the philopatric sex and are thought to have a better knowledge of ecological resources compared to males who disperse[43].

Overall, these analyses provide evidence that social transmission was option specific, i.e., observing lift increased the rate at which observers learned to solve the task using the lift option, and likewise for the pull option (see Table 2 for estimates of effect sizes and total support for each network). This suggests, confirming previous studies[25,26,44,45] that vervet monkeys demonstrate social transmission that could lead to foraging group-level traditions. However, we also found that once an individual had solved the task using one option it became an estimated 31x more likely to asocially learn the other technique than individuals naive to both techniques (95% CI = 11.3–110.6, support = 100.0%, Table 2, see section "Testing whether the learning generalization effect operated on asocial or social learning" In Supplementary Note 2). This suggests that learning one option for obtaining food does not inhibit learning of the other option; rather, that it makes later asocial learning of the other option more likely (see Fig. 3 and Supplementary Fig. 2 in Supplementary Note 2). The probability of learning is 31 times higher when the individuals already learnt one option (Fig. 3). An estimated 45.1% of both learning events (the first and the second learnt options) occurred by OS social transmission (95% CI: 31.0–53.6, Table 2), suggesting most monkeys learned their first option by OS social transmission then rapidly learned the other option by asocial learning (i.e., regardless of their observational experience). All these results taken together show that, despite the fact that individuals learnt both options over the course of the experiment through both social and asocial learning, they had, for the majority of them, a favorite technique, which was the lift

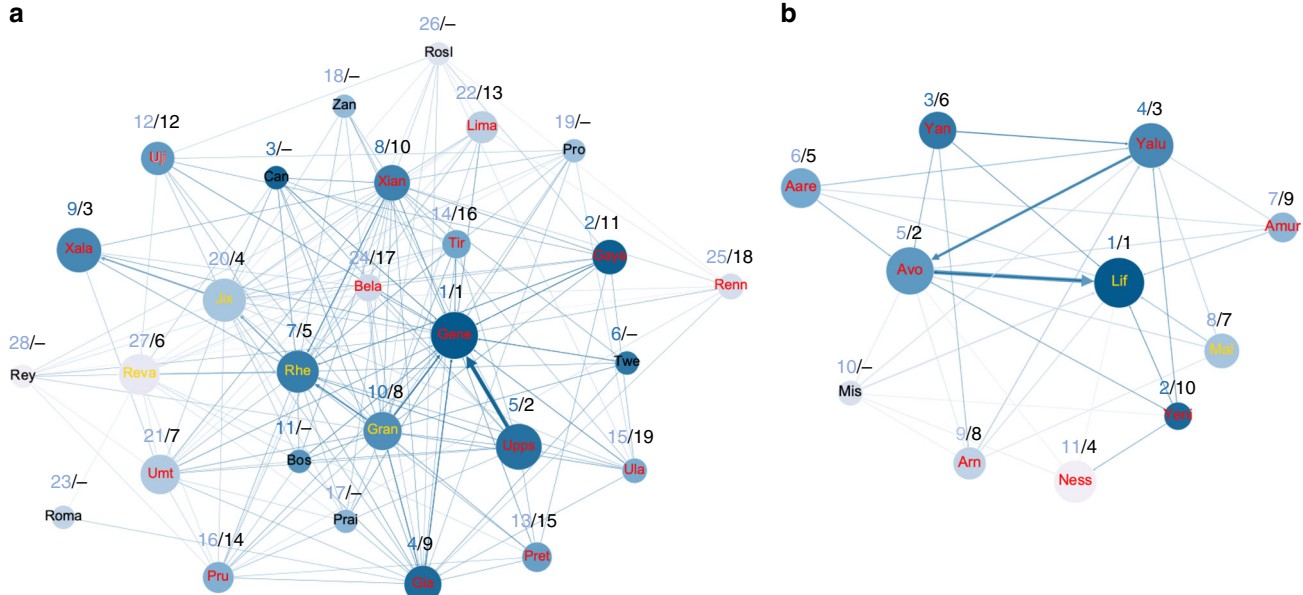

**Fig. 2 Sociograms depicting transmission pathways. a** Transmission pathway in Noha. **b** Transmission pathway in Kubu. Each node represents an individual labeled by its name (three letters code for males, four letters code for females). Individuals whose name is in red first succeeded to use the lift technique; individuals whose name is in yellow first succeeded to use the pull technique; individuals whose name is in black did not succeed the task. The color gradation of the nodes represents the hierarchical ranks: dark blue represents higher-ranked individuals while light blue represents lower-ranked individuals. Numbers written in blue correspond to the ranks. Size node is ranked according to the order of acquisition of the task: bigger is a node, earlier an individual learnt the task. Numbers written in black correspond to the order of acquisition of the task. The sign "-" signifies that the individual did not learn the task. Edges between individuals represent the average rate of observation of an individual by another while naive. The arrow signifies the direction of the observation. The thicker an edge is, the bigger the average rate of observation is.

**Table 2 NBDA aim 1: detecting and quantifying social transmission.**

|  | Model-averaged estimate | 95% CIs | Total akaike weight (%) |
|---|---|---|---|
| *Social transmission per observation relative to baseline asocial learning rate* | | | |
| Option specific ($s_{OS}$) | 0.237 | 0.086–2.00 | 95.1 |
| Cross-option ($s_{CO}$) | 0.019 | 0–1.15 | 33.7 |
| $s_{OS} - s_{CO}$ | 0.218 | 0.077–2.76 | |
| *Estimated % of acquisitions by each pathway* | | | |
| Option specific ($s_{OS}$) | 45.1 | 31.0–53.6 | |
| Cross-option ($s_{CO}$) | 5.6 | 0–18.4 | |
| *Effects on asocial learning* | | | |
| Other option solved | x30.7 | x11.3–x110.6 | 100.0 |
| Option (pull/lift) | x0.557 | x0.16–x1.36 | 56.9 |
| *Effects on social transmission* | | | |
| Sex (female/male) | x2.15 | x1.1–x9.3 | 57.1 |

Model-averaged estimates (MAEs) from a network-based diffusion analysis (NBDA) testing for option-specific social transmission, with 95% confidence intervals (CIs). MAEs for effects on asocial learning and social transmission are only shown for variables with >50% support and are back-transformed to the multiplicative scale. See Supplementary Notes 2 and 3 for estimates of effects with <50% support. In all, 95% CIs were calculated using the profile likelihood method from the best model containing that variable.

option. Except for one individual who kept its initial preference for the pull option, all preferred the lift option during the whole experiment (Supplementary Table 1 in Supplementary Note 1; binomial test: NH group-level preference: $p = 0.003$). This preference for the lift technique could have multiple explanations: perhaps the lift option was easier to use than the pull one, or maybe the participants conformed to the majority of individuals watched, or options watched, or may be because specific individuals preferentially used the lift option. This maintained preference of one option within groups could be at the origin of cultural transmission of new behaviors[2].

Our findings support Heyes' hypothesis[16] that the operation of a high-fidelity transmission mechanism is unlikely to determine

alone the formation of traditions and culture. According to Heyes[16], a high-fidelity transmission mechanism does not necessarily insulate the transmitted behavior from change once it has been added to the observer's repertoire. This is indeed the case in our experiment since some individuals who first learnt the pull technique then preferred the lift option over the course of the whole experiment, showing this flexibility. Our results are also in agreement with recent findings arguing that varying degrees of fidelity in transmission can lead to cumulative cultural evolution in both baboons and human infants[46,47].

**Wild vervet monkeys learn from higher-ranked groupmates.** To test for model-biases (kin versus non-kin; females versus

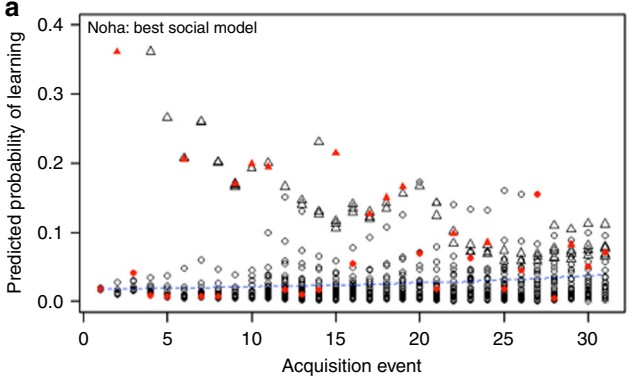
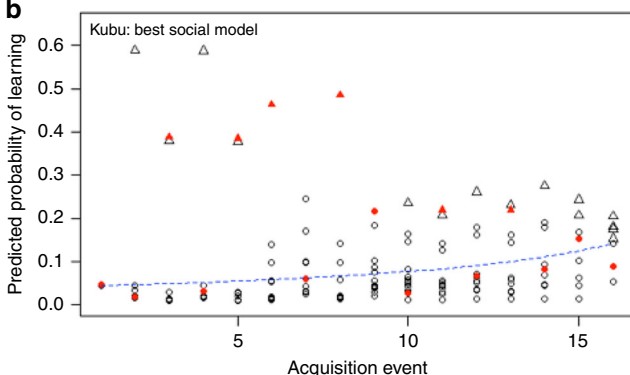

**Fig. 3 Predicted probability for learning in each group. a** NH group and **b** KB group. Each point is an individual x option combination. Triangles represent individuals who have already learned one option. Individuals who learned the task are in red. The blue line represents the average probability of learning across all individuals. The better the model fit, the more red points are above the blue line. Note that triangles tend to be plotted at a high probability, this means that an individual who has learned one option is more likely to learn the other option regardless its observational experience.

**Table 3 NBDA aim 2: identifying the typical pathway of social transmission.**

| Transmission pathways | Total Akaike weight (%) |
|---|---|
| Higher to lower ranks only | 69.8 |
| Higher to lower > lower to higher | 18.8 |
| No bias | 10.8 |
| Lower to higher ranks only | 0.02 |

Support for different pathways of transmission in an NBDA testing for rank biases.

males; adults versus non-adults; high-ranking individuals versus low-ranking ones) we broke the observation network into multiple networks, each scaled by a different $s$ parameter corresponding to the rate of transmission in each pathway[48] (see Methods). For example, to test for the effects of a rank bias we input one network with connections from higher to lower ranking monkeys ($N_{HL}$) and another network with connections from lower to higher ranks ($N_{LH}$). If $s_{HL}$ for network $N_{HL}$ is estimated to be higher than $s_{LH}$ for network $N_{LH}$, then this suggests a dominance bias. We can test for the statistical significance of this bias by comparing the model with a null model ($s_{HL} = s_{LH}$) using AICc. By constructing networks corresponding to the different "from whom" social-learning strategies, NBDA helps us examine which social-learning strategy best predicts the observed pattern.

We found evidence of a rank bias in transmission, with models in which there was only transmission from higher to lower ranks receiving most support (69.8%, Table 3). The relative rate of transmission for this pathway per observation was estimated to be $s_{HL} = 0.23$ (95% CI = 0.020–1.09) compared to $s_{LH} = 0$ (95% CI = 0–0.59) for transmission from lower to higher ranks. While a recent study on vervet monkeys[27] using a different method reported no bias towards neither the dominant females' technique nor towards the lower-ranking ones, our findings suggest that observations of higher-ranked individuals had a greater effect on observers than observations of lower-ranked individuals. There was no evidence for an effect of rank on the rate of being observed (GLMM: estimate = −0.054; standard error = 0.75; $z$-value = −0.72: $p = 0.47$) but an effect of the group with individuals in the smallest group KB observing more others than in the biggest group NH (GLMM: estimate = −1.56; standard error = 0.52; $z$-value = −2.98: $p = 0.0029$). We found an effect on rank and group both on the rate of manipulation and

on the rate of successful box opening. Higher rankers manipulated more than lower rankers (GLM: estimate = -1.55; standard error = 0.70; $z$-value = −2.21: $p = 0.035$), and higher rankers succeeded more to open the boxes than lower rankers (GLM: estimate = −1.43; standard error = 0.61; $z$-value = −2.35: $p = 0.027$). Individuals in KB manipulated more than individuals in NH (GLM: estimate = −1.44; standard error = 0.35; $z$-value = −4.05: $p = 0.0004$) and individuals in KB succeeded more to open the boxes than individuals in NH (GLM: estimate = −1.43; standard error = 0.32; $z$-value = −4.52: $p = 0.0001$). The group effects can be explained by the fact that NH was bigger than KB, leading to more competition for access to the task. These results show that higher-ranked individuals manipulated more and succeeded more to open the boxes compared to lower-ranked individuals but were not more observed than lower-rankers. The single performance bias towards higher rankers does not explain the overall bias from higher rankers to lower rankers because we found evidence of a social information use bias, with observations of higher-ranked individuals having a greater effect on the rate at which behavior is socially learned. This is in accordance with an earlier study[27], in which it was found that the dominant models' demonstrations did not elicit more observations than the low-ranking ones, but interestingly they reported high-ranking females showed a significant bias to copy the dominant female. In that study the authors used a two-option design and trained two models of differing ranks in each group: one was the highest-ranked female and the other one was a female of mid to low rank. A potential limitation of that study is that by imposing the highest-ranked female and a low-mid rank one as models, the lower-ranked observers tested would have the choice between a very dominant model and a less dominant one instead of a dominant and a subordinate one. Moreover, due to the small sampling of models in each group—only two—it is possible that observation of the low-mid rank model had a bigger effect than observation of the highest rank model. In that case, the low-mid rank and highest rank models would not provide a strong inference about low-rank versus high-rank models. In the present study, we believe that such bias is not at work since our analysis allows the inference across numerous higher and lower rankers than just one of each. Furthermore, the earlier study did not include individuals' observations of manipulations by conspecifics that occurred over the course of the experiment, before the observers were tested. Observers could thus have been influenced by the observation of manipulations of other group members—

potentially of higher rank than themselves—before making their first choice of method. By using an open diffusion experiment with no trained model, we mimicked what would arise in a natural situation, and the use of NBDA took into account every observation an individual made before its first success, which represents a more powerful method for identifying the dominance bias we found.

In contrast, we found little evidence of a bias based on kin (support for no bias = 76.2%) or sex (support for no bias = 75.6%) or age (support for no bias = 52.8%, see Supplementary Table 3 and Supplementary Note 3 for support for each hypothesis). While previous studies of vervet monkeys reported both a female bias[25,29] and a mother bias[23,24,26], our findings did not identify any such biases. On the one hand, such inconsistencies could be due to our relatively small sample size compared to previous studies, that tested more than two groups. On the other hand, discrepancies might be explained by the fact that in both cited studies[25,29], female and male models were of high social rank while in our study, they were of varying social ranks. The possibility that different results could have arisen by running the same kind of experiment with only low rankers or low-ranking females and high-ranking males as models is an open question. Finally, in the above cited studies[24,26], only infants of less than one year of age were tested whereas in our study, only individuals of more than one year of age took part in the experiment. It is then possible that young infants focus on maternal figures during a first phase of learning and later widen their attention during a second phase of learning, focusing on specific individuals such as high-rankers[49] who could be considered as experts[49,50].

As NBDA analyzes only the pathways of the initial spread of each behavioral variant, it cannot evaluate which social-learning strategies monkeys use to decide which behavior to settle upon once they have learned both options. Therefore, we could not test with NBDA for a conformist bias as suggested by van de Waal and colleagues[26,51]. In the first of these studies[26], the authors reported that dispersing males adjusted their food color preference to those of their new group, but it was not possible to track exactly what and whom these males observed prior to their preference switching. In the second study[51], the authors found that low-ranked females who split from their natal group expressed a 100% bias for the preferred food color of their original parent group despite prior experience that both options were now palatable. These results have been interpreted as conformity to either the preferences of high-rankers—highly plausible and in accordance with our results—or of a majority in the parent group or both, which remains to be truly tested. One future avenue would be to use another modeling approach such as Experience-weighted attraction models (EWA), which would allow analysis of the entire behavioral sequence of options chosen by each individual. EWA permits examination of which strategy, or combination of strategies, best predict the behavior of individuals conditional upon their personal experience and the unique sequence of events each individual observes in the population[52,53].

## Discussion

Our results offer strong evidence that new box opening techniques spread through social transmission in groups of wild vervet monkeys, consistent with previous findings using two-option designs[27,44]. We found that (i) observing one option increases the probability of learning it, (ii) individuals who learned one option are 31x are more likely to subsequently learn the other option, and (iii) females tended to socially learn twice as fast as males. We reported that the two first solvers were dominants individuals in

both groups and that higher rankers manipulated and succeeded more to open the boxes than lower rankers. However, higher rankers were not more observed than lower rankers; instead, the observation of a higher ranker had more weight than the observation of a lower ranker. We also found an evidence that wild primates may prefer to copy higher-ranked conspecifics for their first success despite the fact that a conformity bias could operate on formation of long-term preferences. Our results build on previous experiments on primate social-learning strategies, in which observational experience was more tightly controlled using a dynamic observation network instead of a static one, by testing for strategies in a more ecologically valid context in which individuals were allowed to freely interact with each other and the task.

We believe that because the strength of our findings results from the fact that we combined innovative, ecologically valid field experiments with a powerful modeling analysis, which is flourishing and promising in the field of social learning. Our findings contribute to a better understanding of social-learning strategies and support the possibility of several biases operating within a single species[2]. Our study also contributes to the debate revolving around cultural evolution by supporting the notion[16,46,47] that high-fidelity transmission is not sufficient to support culture. With more studies and analyses of this kind, we will better be able to understand social-learning strategies and what aspects of cultural transmission are shared or not, between species.

## Methods

**Experimental model and subject details.** Two groups of wild vervet monkeys (*Chlorocebus pygerythrus*) took part in the study: "Noha" (NH) and "Kubu" (KB). NH was composed of 28 individuals (two adult males; 6 adult females; 12 juveniles males; 8 juveniles females; Table 1) and KB was composed of 12 individuals (1 adult male; 6 adult females; 5 juveniles males; Table 1). Males were considered as adults once they dispersed, and females were considered as adults after they gave their first birth. Individuals that did not fulfill these criteria were considered as juveniles[44]. Each group had been habituated to the presence of human observers: since 2010 for NH which had more box experiments than KB[27,44], and since 2013 for KB. All individuals were identifiable thanks to portrait photographs and specific individual body and face features (scars, colors, shape etc.).

Ethics guidelines: our study adhered to the "Guidelines for the use of animals in research" of Association for Study of Animal Behavior and was approved by the relevant local authority, Ezemvelo KZN Wildlife, South Africa.

**Study site.** The study was conducted at the Inkawu Vervet Project (IVP) in a 12,000-hectares private game reserve: Mawana (28°00.327S, 031°12.348E) in KwaZulu Natal province, South Africa. The vegetation of the study site consisted in a savanna characterized by a mosaic of grasslands and clusters of trees of the typical savannah thornveld, bushveld and thicket patches. Mawana houses various species of animals, including elephants, hippopotamus, giraffes, zebras, and numerous species of antelopes. The common predators of vervet monkeys consist in hyenas, jackals, caracals, servals and several species of snakes and raptors.

**Hierarchy establishment.** Agonistic interactions (e.g., stare, displacement, chase, hit, bite) were collected from May 2017 to October 2017 out of experiment days on all the adults and juveniles of both group (excepted one adult female in KB who was not sufficiently habituated to human presence and her offspring from 2016) via ad libitum sampling method[54] and food competition tests (i.e., corn provided to the whole groups from a plastic box). Data were collected by CC and different observers from the IVP team. Before beginning data collection, observers had to pass an inter-observer reliability test with 80% of reliability for each data category between two observers. Data were collected on handheld computers (Palm Zire 22) using Pendragon software version 5.1 and, at the end of the study, on tablets (Vodacom Smart Tab 2) equipped with the Pendragon version 8.

Individual hierarchical ranks were determined by the outcome of dyadic agonistic interactions recorded ad libitum and through food competition tests using Socprog software version 2.7[55]. Hierarchies in both groups were significantly linear (NH: $h' = 0.29$; $P < 0.0001$; KB: $h' = 0.80$; $P < 0.0001$) and ranks were assessed by I&SI method[56].

**Open diffusion experiment.** The experimental apparatus consisted in a two-option design: a transparent plastic box with the back and adjacent sides painted in blue (dimensions: 13 × 10.5 × 7.9 cm) that could be opened in two different ways to gain access to a thin slice of apple inside: a lid on the top of the box could be lifted

(Fig. 1a) and a drawer in the front could be pulled (Fig. 1b). These artificial fruits were new to the monkeys already previously tested on different two-door artificial fruit experiments[27,44].

Experiments took place at sunrise at monkey's sleeping site. Four experimental boxes, spaced by about two meters from each other, were anchored to the ground using camping hooks at two different locations that were spaced by about 20 to 50 meters to prevent monopolization of boxes by a single individual, for a total of eight boxes simultaneously available. CC led the experiment with the help of one or two field assistants so that at least one person was located at one of the two locations. All monkeys were free to interact with the boxes within the constraints of the social group dynamics, such as rank, and to learn by themselves how to open the boxes. Once a set of four boxes was empty, an experimenter approached the boxes and rebaited it. An experimental session stopped when an individual ate the amount of one apple or when the group decided to leave the sleeping site. Experiments were video recorded using a JVC camera (EverioR Quad Proof GZ-R430BE) to which the experimenter said aloud the identities of the actor and of the attending neighbors for each manipulation event. A manipulation event was defined either as an attempt to open one box (i.e., the individual acted on the box failing to fully open it and to get access to the food) or as a success (i.e., the individual succeeded to fully open the box and to remove the food). A conspecific was considered as attending when it had its head or body oriented in an unobstructed line towards the subject manipulating the box and was located within 0–30 m from the boxes. Several individuals could thus be registered as attending to one or several actors simultaneously.

The open diffusion experiments ran from May 2017 to August 2017 to have a maximum of individuals participating to the experiment. Individuals (marked with a "*" in Table 1) who had been recorded as observers, but who did not manipulate the boxes were tested opportunistically during normal field work days with a single box when the individuals who participated the most were not around. The aim of this final step was to test whether these naive observers learnt something from their observations. A total of 17 sessions of open diffusion experiment were run in NH and 12 in KB. The average duration of an experimental session was 81.44 min for NH and 62.01 min for KB.

**Video analysis.** Half of the video recordings were later analyzed by CC and half by a field assistant with Media Player Classic Home Cinema software version 1.7.11. Twenty percent of the video were analyzed by both observers and the inter-observer reliability was 0.87. During video analysis in slow motion or frame by frame, the following variables were encoded: the date, the exact time of each manipulative event, the identity of the actor, the technique used (lift; pull; alternative: return and lift; return and pull) and the identity of attending individuals.

**Quantification and statistical analysis.** Data were analyzed using Network-Based Diffusion Analysis (NBDA) in the R statistical environment v3.5.2[57] using the NBDA package v0.7.10 [58] (scripts are provided in the Supplementary Software). NBDA[39,41,59] assumes that the rate of social transmission between individuals is proportional to the network connection between them and its aim is twofold: (1) to detect and to quantify social transmission and (2) to establish the typical pathways of social transmission. We used a multi-network NBDA[8,48] allowing us to input multiple networks, each scaled by a different s parameter, which estimates the rate of transmission per observation through each network[39]. We used the order of acquisition (OADA) variant of NBDA[40], which takes as data only the order in which individuals acquire the target behavior and not the times of acquisition. Full specifications of all models and additional explanations are given in Supplementary Notes 2 and 3.

Binomial tests (see results in Supplementary Table 1 in Supplementary Note 1) were run to test whether individuals significantly used more one option over the other on the course of the experiment (i.e., test for a preference for one technique over the other after having learnt the task) and group-level preference for NH (not enough data were recorded on all group members in KB) using the "binomial.test" function in R 3.5.2.

Sociograms depicting transmission pathways and observation patterns (Fig. 2a, b) were created with Gephi 0.9.2 software[60]. To build these sociograms, and for the sake of clarity and accuracy, we calculated for each individual, the average rate at which individual A observed individual B when A was still naive. This rate was calculated as: (total number of times A observed B solving the task using lift option prior to A learning lift + total number of times A observed B solving the task using pull option prior to A learning pull)/cumulative solving time for lift and for pull.

The first aim of NBDA is to detect and quantify social transmission. The order of acquisition of each technique (lift; pull) was determined based on the date and time when individuals succeeded in opening the box for the first time using one of these two techniques (henceforth "options"). Here, we aimed to assess whether social learning was occurring and whether its effects were option-specific or generalized between the two options. A dynamic social network $o_{ijl}(t)$ was created from open diffusion experiment data reflecting the number of times $i$ observed $j$ performing option $l$ (1 = lift; 2 = pull) prior to time $t$. We included networks representing two different pathways of learning: option-specific social learning and cross-option social learning (see Supplementary Note 3 for further details). Option-specific (OS) social learning occurs when observation of a task solution using the lift option increases the rate at which the observer learns the lift option, and

likewise for the pull option. Conversely, cross-option (CO) social learning occurs when observation of a task solution using the lift option increases the rate at which the observer learns the pull option, and vice versa. In a standard NBDA the s parameter estimates the rate of social transmission per unit connection relative to asocial learning. In our model each network has an associated s parameter, denoted $s_{OS}$ and $s_{CO}$.

We controlled for the possibility that vervet monkeys learn one option more easily by asocial learning by including "option" as a factor influencing the rate of asocial learning. We also allowed for the possibility that vervet monkeys might generalize their learning, i.e., learning to solve the task using one option might increase the rate at which they subsequently learned the other option by asocial learning. Alternatively, learning one option might inhibit learning of the other, which would reinforce formation of group-level traditions.

Sex, age class (adult/non-adult), and rank were included as individual level variables (ILVs) that potentially influence the task solving order. All variables were standardized so they were centered on zero, with a range of 1. We used the "unconstrained" model to include the effects of ILVs, which independently estimates the effects each ILV has on asocial and social learning[2]. We used a multi-model inference approach using Akaike's Information Criterion corrected for sample size (AICc)[61] to obtain the support for each of the following hypotheses: (a) a different rate for each pathway ($s_{OS} \neq s_{CO}$); (b) option-general social transmission ($s_{OS} = s_{CO}$); (c) OS social learning only ($s_{CO} = 0$); (d) CO social learning only ($s_{OS} = 0$); and (e) asocial learning ($s_{OS} = s_{CO} = 0$).

For each of a-d we fit models with every combination of five ILVs affecting asocial learning and 3 ILVs affecting social learning, resulting in 256 models for each set. For (e) asocial learning, ILVs can only affect asocial learning, resulting in only 32 models. We calculated the total Akaike weight as a measure of support for each hypothesis a-d[61]. Owing to the lower number of models in the asocial set (e) we do not use the total Akaike weight as a measure of support for asocial learning, instead we use the 95% confidence intervals for the s parameters to this end.

The second aim of NBDA is to identify the typical pathway of social transmission. We extended the NBDA model described above to test for social information use biases in the transmission pathways. Since there was strong support for OS social learning only, we simplified the model by dropping the CO effect. The biases we tested for were as follows:

Rank biases. Does transmission rate from higher to lower-ranked vervet monkeys differ from that from lower to higher ranks?

Sex biases. Does the rate of transmission differ between male and female transmitters?

Age biases. Does the rate of transmission differ among and within different age classes?

Kin biases. Does the rate of transmission differ between kin and non-kin, and between different classes of kin (mother to offspring, offspring to mother, between siblings)?

We partitioned the dynamic observation network into mutually exclusive networks, e.g., for rank, connections from higher to lower ranks, and connections from lower to higher ranks. This enable us to compare models in which there was a transmission bias (different s parameters for each network), models in which there was no transmission bias (equal s parameters), and transmission only in each pathway (relevant s parameter constrained to zero). In each case we fitted 256 models with every combination of ILV effects, and compared total Akaike weights for each hypothesis, as described above. Evidence of a transmission bias would indicate that there was a stronger effect per observation in one pathway than the other. See the Supplementary Note 3 for further details of how we partitioned the network in each case.

We ran the whole NBDA analysis as above on pooling the "return and lift" (srtli) with "lift" and "return and pull" (srtpu) and "pull" successes instead of only "lift" (sli) and "pull" (spu) to test whether we obtained different results (see Supplementary Fig. 3 in Supplementary Note 2; Supplementary Table 3, 4; Supplementary Table 5 in Supplementary Note 3). We obtained the same results than when focusing on "li" and "pu" techniques, this is why only NBDA on these techniques is presented in the present paper.

To test for a ranked-based performance bias two generalized linear models (GLMs) (quasi-Poisson error and log link function) were used to test the effect of rank and group on (i) the success of opening the boxes (using lift or pull) and on (ii) the rate of manipulation by informed individuals (attempts and successes). We added the log of the time each individual had to manipulate the task once they had first solved it as an offset (a standard statistical technique for converting a Poisson GLM for analysis of counts into a model for analyzing counts per unit of time). A generalized linear mixed model (GLMM) (Poisson error and log link function) was used to test the effect of rank and group on the frequency of observations between members of each dyad. This model tests for a combined observation x performance bias. In the GLMM, rank and group were considered as fixed effects, the identity of observer and observed were included as random effects and the log of the time each individual had after its first success was added as an offset. All tests were performed with package lme4.

**Reporting summary.** Further information on research design is available in the Nature Research Reporting Summary linked to this article.

## Data availability
All data are included in Supplementary Data 1-15.

## Code availability
The code used for the analysis is included in Supplementary Software 1–5.

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

## Acknowledgements

We thank Arend van Blerk and the whole IVP team for their help and support in the field. We warmly thank Justine Mertz for her help during the experiment and for video coding. We are also particularly thankful to David Lemieux, Marie-Laure Poiret, Karin Snyder, Annaëlle Surreault, Alysson Couchouron, Rachel Arnaud, Shea Hamilton, and Hermine Saint-Jean for their assistance in data collection and during the experiment. We are grateful to the van der Walt family for their permission to conduct the study on their land. We greatly thank Matthias Wubs for writing some R scripts for us to extract social data as matrixes. We are grateful to Andrew Whiten for providing us with the boxes and for his feedback on the manuscript. We are thankful to Cédric Sueur for his advice in the use of Gephi software. C.C. was funded by postdoctoral fellowships from the Fyssen Foundation and the Fondation des Treilles and IVP was funded by the Swiss National Science Foundation (31003A_159587 and PP03P3_170624) and the Branco Weiss Fellowship—Society in Science granted to E.v.d.W.

## Author contributions

C.C. and E.v.d.W. designed the experiment. C.C. conducted the experiments and analyzed the videos. C.C. and W.H. ran the statistical analyses. C.C wrote the first draft, and W.H. wrote the NBDA scripts, the methods part on NBDA, and the main parts of the supplementary information. C.C., E.v.d.W., and W.H. wrote the second draft.

## Competing interests

The authors declare no competing interests.
