## [Peer Review File · Nature Communications]

Reviewers' Comments:

Reviewer #1:

Remarks to the Author:

Canteloup and coworkers present a study on social learning dynamics in a group of wild vervet monkeys. Their study is based on an open diffusion experiment in which individuals learn to open artificial foraging boxes. To analyze their data the authors use a variant of network-based diffusion analysis (NBDA). Besides testing for the occurrence of social learning the method allowed testing for various details of the learning processes, including tests for social learning strategies. Thus, the authors found for example that social learning dynamics in this experiment were consistent with a social learning strategy to preferentially copy higher ranking individuals. The presented method and results are particularly valuable for understanding social learning dynamics in wild animals and how individual-level processes translate into group-level cultural dynamics.

In general I liked this manuscript very much. However, for the framework of the study it could be useful to explicitly consider additional individual-level processes that could influence group-level dynamics. In their current conceptual framework the authors included (1) social network structures (L49-54), (2) social learning mechanisms, i.e. the cognitive mechanisms underlying the process of social learning (L55-58), and social learning strategies, i.e. learning biases based on the content of behavior or its context. (L58-67). I think there are at least two additional processes that could influence group-level cultural dynamics: (1) attention biases that are not captured by social networks of shared space use. E.g. an individual could spend equal time with low and high ranking individuals but pay more attention towards what higher ranking individuals are doing, which could create a learning bias that is not captured by social network structure or social learning strategies (at least not the kind of strategies that are inferred in this study). (2) performance biases, where some individuals would consistently perform a certain task more often and thus could serve more often as a model for others. E.g. after learning a behavior, higher ranking individuals might perform behaviors more often compared to lower ranking individuals (which could occur even if monopolization is not possible). Again this could generate a bias towards learning from higher ranking individuals that is not determined by social network structure or social learning strategies.

Considering these additional biases might provide a more solid conceptual foundation for the current study. In addition, a discussion of these additional biases would help a lot to better understand the applied version of NBDA. The way the input networks were constructed, the method automatically controls for any biases generated by social structure and potential attention and performance biases. This is because edges in the networks were only added if a naïve individual was in a position of closely observing a model performing a new behavior. This is very different from a lot previous applications of NBDA in which static networks were used. In these versions, it was not possible to control for potential attention and performance biases. I think it could be particularly important that readers understand such differences to correctly interpret the results and correctly apply this methods in future studies.

Regarding the interpretation of the results it might be worth mentioning that not investigated attention or performance biases could enhance or weaken the inferred the learning bias of preferentially copying higher ranking individuals. Perhaps such additional biases could also explain the discussed differences to previous studies? Ideally the conducted study would be complemented by analyzing different kinds of biases, or the combined effect of all biases. However, such an extension might be beyond the scope of the current study.

Minor comments:

L182 and L188: Non-significant results should not be taken as evidence for the absence of an effect. I suggest rewording to something like "no statistically significant preferences could be detected". In

addition, I suggest to show the actual p-value instead of writing " $p > 0.05$ ".

L373: It would be very useful to clarify what a "manipulation event" is. Is it just the manipulation of the box or is it a successful manipulation, where the individual succeeded in opening the box? I guess the latter (see also L413), but it is not entirely clear.

L577: the journal should be PLoS Biology

Reviewer #2:

Remarks to the Author:

This paper presents a study on social learning in wild vervet monkeys that combines an experimental with a modelling approach based on network based diffusion analysis (NBDA). Two groups of wild vervet monkeys were first tested in a foraging task in which they were confronted with two sets of 8 artificial fruits each. Each fruit could be opened in two ways, either by lifting a lid or pulling a drawer, to access a food reward. The empirical question was whether a given subject would be affected by the technique that other group members had employed when the subject was still naive, i.e. it had not succeeded to open one of the artificial fruits before. The NBDA was used to contrast different modes of transmission in the group, specifically option specific (OS) learning vs. cross-option (CO) social learning. The best fit was found for the OS model. Furthermore, an effect of rank on social learning was detected, with subjects being more likely to learn the opening of the box when they had observed higher-ranking subjects first. This is an interesting approach, but I have a number of concerns.

- I am not convinced that copying indeed takes place. To demonstrate this, the authors should report very clearly in the text how many subjects who had only seen ONE technique then actually used this technique. Indeed, an inspection of the data from Noha shows the following sequence of events: Lif used the Drawer (apu=pull[?; legend is missing]), but Avo, who was attending, used lift. The next animal also used lift and had observed Avo using lift, but the next animal (Ness) used "srtpu (pull?)" and had just observed both options. Given that the group has only 12 subjects, out of which only 10 learned to open the box, and discounting the first subjects that obviously did not learn to open the box socially, this leaves N=9 subjects for which I have doubts for 2 already. I did not go through the rest of the data. On a side note, remember that all data need to be annotated, i.e. all states of a variable need to be explained.
- Alternatively, if the authors define social learning as using the technique that the vervets have observed more frequently than the other one before, then the problem arises that the outcome becomes hard to distinguish from a simple model that combines priority-of-access, i.e. dominants have access first, with a general bias for lift. In other words, the strong bias for the lift option may create the artefact that the animals are doing what the others are doing, while they may simply all largely prefer one option.
- If the authors can rule out the problems mentioned above about social learning, they should consider the possibility that the rank effect may simply be due to the fact that subjects are more likely to "learn" from dominants because these have access the boxes first and therefore it is more likely that they have been observed.
- What is the theoretical background that gives rise to Cross-Option learning? This appears to be a bit of a straw-man ... Why would a subject be more likely to do the OPPOSITE of what they have observed?
- Much more descriptive information on the actual behavior needs to be given: what is the latency to approach once the first subject has opened the box; what is the distance at which it is reasonable to assume that subjects are attending. How many subjects typically observed another one? Were there any cases where two subjects were available as models, and if so, how was this coded? What if the animals used different techniques?
- The sample size is small (only 2 groups and few subjects in one group). I therefore strongly suggest to simulate different data sets that reflect different transmission processes for groups of different size

and composition and check the outcome. For instance, what if the all subjects learn to open the box individually, but have a bias of using the lift option. Would the outcome of the diffusion analysis yield a similar pattern or could the two processes be distinguished?

- It is worrying that one of the outcomes of the model is that "After having learnt one option, individuals were 31x faster to subsequently asocially learn the other option [...]" (abstracts and a number of places in the ms). Firstly, faster THAN WHAT? And secondly, what are the time scales? Are the authors implying that animals first learn socially to use one SPECIFIC option, in e.g. 2 minutes, and then it takes them only e.g. 4 seconds to discover the other option individually? More information on the actual data is needed to convince the reader that the network diffusion model actually fits the data.
- I don't think it is necessary to invoke "culture" in this transmission experiment. Too much hyperbole here. "Social diffusion" is just fine. But as I said above, it is not really clear what the animals are learning from each other, and which parts (if any) of the observed behavior are facilitated by the presence of others.
- All terms need to be clearly defined. It is well known that people have different ideas what 'copying' means, etc. E.g. Tomasello distinguishes between imitation and copying; others don't (line 55 ff).

Minor comments

Line 25: twice as fast

Line 85: this study does not address social innovation, cut these parts, they are distracting.

97: what is the difference between ii) and iii)?

102: specify directed social learning

104 ff: too vague – use the section to develop predictions

156: explain what the number means in the text

163: compared, not compare

168: "capable of social transmission that could support ..." odd wording, rewrite

341: "not enough" ◊ "not sufficiently"

371: "thanks to" ◊ "using a"

373: distance?

378: "have then been" ◊ "were"

379: "who are participated the most": please specify

406: why individual analyses? Why not at the group level?

536: "The thicker an edge is, the bigger ..."

I have two version of Figure 2, one has orange names, the other does not. Orange on green is bad for people who are color blind.

Reviewers' comments:

Reviewer #1 (Remarks to the Author):

Canteloup and coworkers present a study on social learning dynamics in a group of wild vervet monkeys. Their study is based on an open diffusion experiment in which individuals learn to open artificial foraging boxes. To analyze their data the authors use a variant of network-based diffusion analysis (NBDA). Besides testing for the occurrence of social learning the method allowed testing for various details of the learning processes, including tests for social learning strategies. Thus, the authors found for example that social learning dynamics in this experiment were consistent with a social learning strategy to preferentially copy higher ranking individuals. The presented method and results are particularly valuable for understanding social learning dynamics in wild animals and how individual-level processes translate into group-level cultural dynamics.

In general I liked this manuscript very much. However, for the framework of the study it could be useful to explicitly consider additional individual-level processes that could influence group-level dynamics. In their current

conceptual framework the authors included (1) social network structures (L49-54), (2) social learning mechanisms, i.e. the cognitive mechanisms underlying the process of social learning (L55-58), and social learning strategies, i.e. learning biases based on the content of behavior or its context. (L58-67). I think there are at least two additional processes that could influence group-level cultural dynamics: (1) attention biases that are not captured by social networks of shared space use. E.g. an individual could spend equal time with low and high ranking individuals but pay more attention towards what higher ranking individuals are doing, which could create a learning bias that is not captured by social network structure or social learning strategies (at least not the kind of strategies that are inferred in this study). (2) performance biases, where some individuals would consistently perform a certain task more often and thus could serve more often as a model for others. E.g. after learning a behavior, higher ranking individuals might perform behaviors more often compared to lower ranking individuals (which could occur even if monopolization is not possible). Again this could generate a bias towards learning from higher ranking individuals that is not determined by social network structure or social learning strategies.

Considering these additional biases might provide a more solid conceptual foundation for the current study. In addition, a discussion of these additional biases would help a lot to better understand the applied version of NBDA. The way the input networks were constructed, the method automatically controls for any biases generated by social structure and potential attention and performance biases. This is because edges in the networks were only added if a naïve individual was in a position of closely observing a model performing a new behavior. This is very different from a lot previous applications of NBDA in which static networks were used. In these versions, it was not possible to control for potential attention and performance biases. I think it could be particularly important that readers understand such differences to correctly interpret the results and correctly apply this methods in future studies.

Regarding the interpretation of the results it might be worth mentioning that not investigated attention or performance biases could enhance or weaken the inferred the learning bias of preferentially copying higher ranking individuals. Perhaps such additional biases could also explain the discussed differences to previous studies? Ideally the conducted study would be complemented by analyzing different kinds of biases, or the combined effect of all biases. However, such an extension might be beyond the scope of the current study.

Thank you for your positive review. We agree with you on these two points: attention and performance biases are two additional processes that could indeed influence social learning strategies. We have added a section (l121-137) which explains how an overall bias could be a result of any combination of these three component biases. We have made it clear that our primary focus is on what we term a “social information use bias”. However, in addition we made some attempt to assess the other two biases in this case, in reference to rank. We ran GLMs and GLMM and found that higher rankers significantly manipulated and succeeded more to open the boxes than lower rankers. However, we found no effect of rank on the probability of being observed, i.e. higher rankers were not more observed than lower rankers. Thus, there is no indication of a performance bias towards higher rankers that would enhance the rate at which higher rankers are observed. This result is in line with the overall effect of the observed social information use bias we found with NBDA. We added these results in the results and discussion sections and we discussed these potential biases in the discussion of our manuscript (l 350-370). We also explained further the motivation for to using a dynamic observation network *versus* a static one in NBDA (l 163-175).

Minor comments:

L182 and L188: Non-significant results should not be taken as evidence for the absence of an effect. I suggest rewording to something like “no statistically significant preferences could be detected”. In addition, I suggest to show the actual p-value instead of writing “ $p>0.05$ ”.

We agree and corrected these sentences and displayed the actual p-values in the text (l 264; l 280).

L373: It would be very useful to clarify what a “manipulation event” is. Is it just the manipulation of the box or is it a successful manipulation, where the individual succeeded in opening the box? I guess the latter (see also L413), but it is not entirely clear.

A manipulation event is either an attempt to open the box or a success to open the box. We added precise definitions in the text (l 506-509).

L577: the journal should be PLoS Biology

Thank you for this correction, we fixed the typo.

Reviewer #2 (Remarks to the Author):

This paper presents a study on social learning in wild vervet monkeys that combines an experimental with a modelling approach based on network based diffusion analysis (NBDA). Two groups of wild vervet monkeys were first tested in a foraging task in which they were confronted with two sets of 8 artificial fruits each. Each fruit could be opened in two ways, either by lifting a lid or pulling a drawer, to access a food reward. The empirical question was whether a given subject would be affected by the technique that other group members had employed when the subject was still naive, i.e. it had not succeeded to open one of the artificial fruits before. The NBDA was used to contrast different modes of transmission in the group, specifically option specific (OS) learning vs. cross-option (CO) social learning. The best fit was found for the OS model. Furthermore, an effect of rank on social learning was detected, with subjects being more likely to learn the opening of the box when they had observed higher-ranking subjects first. This is an interesting approach, but I have a number of concerns.

Thanks for your review and for having raised all these interesting points. We have made changes to the paper to address each point and provide details in our responses to each point below.

- I am not convinced that copying indeed takes place. To demonstrate this, the authors should report very clearly in the text how many subjects who had only seen ONE technique then actually used this technique. Indeed, an inspection of the data from Noha shows the following sequence of events: Lif used the Drawer (apu=pull[?; legend is missing]), but Avo, who was attending, used lift. The next animal also used lift and had observed Avo using lift, but the next animal (Ness) used "srtpu (pull?)" and had just observed both options. Given that the group has only 12 subjects, out of which only 10 learned to open the box, and discounting the first subjects that obviously did not learn to open the box socially, this leaves N=9 subjects for which I have doubts for 2 already. I did not go through the rest of the data.

We believe it would only be necessary to show this if we were claiming that *all* vervets (save the innovators) learned by social transmission (copying), and/or that observation *always* leads to learning. These, we believe would be straw-man versions of the social transmission hypothesis. We are assuming that at least some individuals learned by asocial learning, as indeed they must, since there were no trained demonstrators. We are aiming to quantify the strength of social transmission- as estimated by the extent to which the rate of learning of each option is increased by observations of that option. This we also express as an estimate of the proportion of individuals that learned by social transmission. Therefore, it is not surprising that some individuals would learn an option without first having seen it, nor does it necessarily follow that an individual that observes an option must go on to learn it, if social transmission is operating. The NBDA model quantifies the strength of evidence for social transmission based on how well the number of observations of each option by each individual, predicts the pattern of spread. Thus, our analysis already accounts for the information of how many individuals had seen an option prior to learning it, when estimating the proportion of individuals that learned by social learning.

On a side note, remember that all data need to be annotated, i.e. all states of a variable need to be explained.

We apologize for forgetting to add a legend file for the raw data file. This is now provided (cf. 'legend for raw data file.txt').

- Alternatively, if the authors define social learning as using the technique that the vervets have observed more frequently than the other one before, then the problem arises that the outcome becomes hard to distinguish from a simple model that combines priority-of-access, i.e. dominants have access first, with a general bias for lift. In other words, the strong bias for the lift option may create the artefact that the animals are doing what the others are doing, while they may simply all largely prefer one option.

The model of social learning underlying the NBDA assumes that the rate at which vervets learned each option is proportional to the number of times they have seen it, rather than that vervets will *always* use the option they have seen performed most. The option specific effect we report is highly unlikely to be a spurious result of a confounding bias towards the lift option, since a variable allowing for possible differences in the rate of asocial learning is included in the analysis. Hoppitt, Boogert and Laland (Ref. 40) showed in a general case that inclusion of potentially confounding variables in an NBDA can statistically control for their effects and prevent spurious social transmission effects being detected.

To be certain that this applies in the specific case highlighted by the reviewer (that an asocial option bias might result in a spurious option specific effect), we ran some simulations in which the observed bias towards lift was assumed to be entirely due to an asocial bias. We simulated the option choice based only on this bias. We retained the same pattern of observations but, in our simulations, these had no effect on option choice. We then fitted the same model that was favoured in the real data (one in which social transmission was option specific

only), and the equivalent model of option-general learning. We also fitted another two models in which an asocial bias for one or other option was added- to statistically control for its effects as we did in the real analysis. We then recorded the AICc for the best option general model minus the AICc for the best option specific model as measure of evidence for an option specific effect in the simulated dataset. We then repeated this process 1000 times, and calculated the proportion of simulations in which the evidence for option specific social transmission exceeded that observed in the real data, and found it to be 0.0475- thus confirming that our finding for option specific social transmission is unlikely to be a result of a preference for one option (cf. see script in SI).

- If the authors can rule out the problems mentioned above about social learning, they should consider the possibility that the rank effect may simply be due to the fact that subjects are more likely to “learn” from dominants because these have access the boxes first and therefore it is more likely that they have been observed.

This effect, if it occurred, could not account for the rank effect we report. We find evidence that, *per observation*, observations of higher-ranking individuals have more effect than observations of lower ranking individuals. i.e. we show evidence that *each observation* of higher-ranking individuals carries more weight than observations of lower ranking individuals. Therefore, this effect cannot simply be a result of more performances by, and thus more observations of higher-ranking individuals. We have edited the manuscript to make this distinction clearer, inserting a new paragraph (l 97-113) to break down the different types of bias and explaining where our analysis fits in.

- What is the theoretical background that gives rise to Cross-Option learning? This appears to bit of a straw-man ... Why would a subject be more likely to do the OPPOSITE of what they have observed?

We agree that the model in which subjects learn *only* to do the opposite of what has been observed (i.e where the OS effect, Sos , is constrained to 0) is, *a priori*, extremely unlikely, and just included for logical completeness, and allows us to empirically support the implausibility of the model. (It would be a straw man if it were our key null hypothesis, but it is not- see below).

However, this is not the case for models in which the cross-option (CO) effect is included *alongside* the option-specific (OS) effect, these represent an *a priori* highly plausible case:

First, we have the models in which there is an option-specific effect and a cross-option effect, with effect sizes not constrained to be the same. Whilst the CO effect (Sco) is not *a priori* constrained to equal to or less than the OS effect (Sos), this is what we would anticipate and indeed what we empirically find in these models. When $Sos >$ or $= Sco$ these models represent the case in which the social transmission effect generalizes to some extent, to the other option, with the difference $Sos - Sco$ estimating the extent to which social transmission is option specific. This could be a result of a large scale local enhancement effect, which attracts observers to the task (thus making them more likely than they were before to learn either option), on top of which is superimposed a more spatially specific effect which makes them more likely to learn the specific option they have observed.

Second, we have the case where $Sos = Sco$ - this represents the case where social transmission generalizes completely between the options. This could be the result of only a large scale local or stimulus enhancement effect that acts to attract observers to the interact with the task. This is vital as null hypothesis against which to test the case above ($Sos > Sco$) and $Sos > 0$, $Sco = 0$ in order to establish the strength of evidence for option specific social transmission.

We have clarified this distinction between the models in the main text on lines 225-233 and referred the reader to the SI for more details.

- Much more descriptive information on the actual behavior needs to be given: what is the latency to approach once the first subject has opened the box; what is the distance at which it is reasonable to assume that subjects are attending. How many subjects typically observed another one? Were there any cases where two subjects were available as models, and if so, how was this coded? What if the animals used different techniques?

We added such precisions in the text (l 186-190; l 510-512) and in Table 1. If two actors were manipulating different boxes at the same time and if the attending individual was oriented and looking in the direction of the two actors, this individual was recorded as attending to these two actors. As you can see in the raw data files, one line corresponds to one manipulation event by one actor and as many as attending can be recorded for one watched event. Because the exact time was recorded each time, it is possible that two events occurred at the same time. Moreover, because four boxes were separated by about 2m from each other, it is reasonably possible that an attending individual located at 10m from the set of boxes could easily watch the options used by two actors acting on two different boxes at the same time. This is not a problem for the dynamic version of NBDA we used, and this is obviously taken into account in the analysis. Two main techniques – lift and pull – were used by monkeys but,

sometimes some individuals used these techniques by returning the box from the back to the front of the box – what we called ‘return and lift’ and ‘return and pull’ (see I 581 and see also legend file). For this reason, we ran the whole NBDA analysis on pooling the ‘li+rtli’ and ‘pu+rtpu’ techniques to check if we got different results (see Fig. S2; Tables S3; S4 in SI). We obtained the same results than when focusing on ‘li’ and ‘pu’ techniques, this is why only NBDA on these techniques is presented in the present paper.

- The sample size is small (only 2 groups and few subjects in one group). I therefore strongly suggest to simulate different data sets that reflect different transmission processes for groups of different size and composition and check the outcome. For instance, what if the all subjects learn to open the box individually, but have a bias of using the lift option. Would the outcome of the diffusion analysis yield a similar pattern or could the two processes be distinguished?

Two groups appear to be a small sample size at first sight, but it would have been way more complicated and less realistic to run the same experiment by collecting the same quality of data on bigger groups for the following reasons. As it has rarely been done before, we recorded a dynamic observation network during the experiment which means that, for each manipulation event of every actor, we recorded online who was attending. This required from two to four people recording these high-quality data during each experiment in the field. As a consequence, if we would have run this experiment in a group of 60 monkeys for example, this would mean that we should use more than 8 boxes simultaneously to avoid monopolization by high rankers, so more video camera and more manpower in the field. Besides, this would have represented a drastically more demanding video analysis work. The video clips of our experiment have been analyzed in slow motion, and even at some point frame by frame, by two coders, which already represent more than 40 hours of fine-grained video analysis. We acknowledge that it would have been ideal to run this experiment in more groups or in bigger groups, but we believe that the combination of such ecologically valid field experiments with the use of powerful modelling approach represents a trade-off between research interest and field reality.

The NBDA technique accounts for the number of individuals in the group, as well as the number that solved the task, when calculating the 95% confidence intervals for parameters and the AICc (Akaike’s information criterion corrected for sample size). Note that NBDA is not a new technique devised for this paper, being now 10 years old, and has now been used extensively within the field, with a number of papers examining the performance and validity of the technique. Extensive simulations of the kind described by the reviewer have been run and published (Hoppitt, Boogert, Laland 2010, Whalen and Hoppitt 2015, Hoppitt 2017) showing that in a large range of scenarios the 95% CIs correctly reflect the uncertainty in the data (i.e. contain the true value of s parameters 95% of the time) and AICCs perform well in recovering the correct model. Thus, these simulations show that a small sample size will only act to reduce statistical power to detect social transmission, not to inflate the false positive error rate for detection of social transmission. It has also been shown that the statistical power of the technique can be high in small groups if the social network is well resolved. Therefore, we feel that simulations investigating the general validity and performance of NBDA are unnecessary and beyond the scope of this paper.

However, we agree that the more specific situation modelled in this paper has not been investigated fully, and we have run the simulation described by the reviewer, in which there is no social transmission and an option bias in asocial learning and find that an option specific effect is only falsely detected only 4.75% of the time (see also our response to the previous comment).

- It is worrying that one of the outcomes of the model is that “After having learnt one option, individuals were 31x faster to subsequently asocially learn the other option [...]” (abstracts and a number of places in the ms). Firstly, faster THAN WHAT? And secondly, what are the time scales? Are the authors implying that animals first learn socially to use one SPECIFIC option, in e.g. 2 minutes, and then it takes them only e.g. 4 seconds to discover the other option individually?

More information on the actual data is needed to convince the reader that the network diffusion model actually fits the data.

We have now specified “than individuals naïve to both techniques”. There is no time scale here, since we are talking about the ratio of two rates. As an example, if one factory produces cars three times as fast as another, it does not require one to specify whether it does so per day, per week, per month or per year, it is the same thing. The reviewer is correct in thinking the implied situation is implausible, due to an omission in the sentence: we have now specified “an estimated 31x faster to **asocially** learn the other technique”. The situation we intended to imply is one in which vervets are very likely to learn their first technique by social transmission, but once they do so they are highly likely to learn the second technique regardless of whether they observe it first- this situation is now described in the following modified sentence as: “suggesting most monkeys learned **their first** option by OS social transmission then rapidly learned the other option by asocial learning (i.e. regardless of their observational experience)”. We hope these corrections have clarified the situation.

• I don't think it is necessary to invoke "culture" in this transmission experiment. Too much hyperbole here. "Social diffusion" is just fine. But as I said above, it is not really clear what the animals are learning from each other, and which parts (if any) of the observed behavior are facilitated by the presence of others.

We agree that it is not necessary to invoke culture as a finding of the current experiment and did not intend to do so. We have removed the word culture from the title to avoid this misunderstanding. However, the topic of animal culture is nonetheless an important part of our aims. While definitions of culture vary widely among fields and even researchers in the same field, a feature of almost all is that culture is reliant on socially transmitted behavior that results in group specific behavior that varies among groups. Thus a key question is why social transmission, which seems common in the animal kingdom, does not result in group specific behavior patterns more often? As we discuss in the paper, some have suggested that it is to do with high fidelity copying mechanisms, others have suggested it has more to do with the insulation of behavior from modification once it has been learned. Our results support the latter in this case- the reason why a group specific behavior pattern did not arise in our experiment is not because copying was not of sufficiently high fidelity, rather it is because once learned the behavior was modified (the other option learned asocially). Thus we maintain our results are relevant to an understanding of animal culture. We do not think we are engaging in hyperbole, since we do not claim to be demonstrating culture in our experiment. Rather we are claiming to have broken down the transmission process in a way that reveals why group specific behavior did not emerge, and thus shed light on the reasons why animal culture is rare, and the conditions that determine when it is likely to emerge.

• All terms need to be clearly defined. It is well known that people have different ideas what 'copying' means, etc. E.g. Tomasello distinguishes between imitation and copying; others don't (line 55 ff).

We used 'copying' to refer to any social learning resulting in matching behaviour, as is common in studies focusing on social learning strategies as opposed to underlying psychological mechanism. We have clarified our use of the term (l 75-84)., referencing Kendal and collaborators recent review on social learning strategies in support of this terminology use (Ref. n°4 in the reference list).

Minor comments

Thanks for these minor corrections that we took into account.

Line 25: twice as fast

Line 85: this study does not address social innovation, cut these parts, they are distracting.

We acknowledge that innovation refers to the invention of a new behavior and that in our experiment, the two box opening techniques were not invented - because already existing - but discovered by monkeys. In that line, we changed "innovator" by "solver" (l 120).

97: what is the difference between ii) and iii)?

ii) and iii) are two distinct questions:

Question ii) aims to test whether an option specific social transmission pathway is at work. It could be that observing one option makes monkeys a) more likely to socially learn that same option, in our analysis this represented by $Sos > Sco$, or b) it could be that it generalizes such that monkeys are equally likely to learn both by social transmission, represented in the model as $Sos = Sco$.

Question iii) aims to test whether monkeys are a) less likely to learn the other option, represented in the model by a negative effect of "otherOption" or b) more likely to learn the other option, represented in the model by a positive effect of "otherOption" or c) no effect, represented by a coefficient for "otherOption" = 0 once they have learned one option.

This is clarified in this way when we introduce the model later in the manuscript. To help the reader understand the difference earlier in the paper we have included some examples on lines 83-91, showing how copying fidelity versus insulation from asocial learning relate to the context of learning a foraging skill.

102: specify directed social learning

104 ff: too vague – use the section to develop predictions

We specified the predictions (l 144-145).

156: explain what the number means in the text

Aikake weight corresponds to the support for each model. In that case, when we write “There was strongest support for exclusively OS social transmission (63.8%) followed by models in which OS social transmission was stronger than CO social transmission (22.2%; see Fig. S1)”, this means that there is 63.8% of chance that the best model is the one with only OS social transmission and 22.2% of chance that the best model is one in which OS>CO social transmission. We added a precision in the text (l 234-235). Note that this is also specified in the SI (l 111-127 of SI file).

163: compared, not compare

168: “capable of social transmission that could support ...” odd wording, rewrite

We rephrased this sentence as follow: “this suggests, confirming previous studies that vervet monkeys demonstrate social transmission that could lead to foraging group-level traditions.” (l 248-249).

341: “not enough” ◊ “not sufficiently”

371: “thanks to” ◊ “using a”

373: distance?

There was no strict distance rule for the definition of an attending individual, but such individual could be within 0-30m from the boxes. We acknowledge that an individual with its head and/or body and head oriented in an unobstructed line towards the subject manipulating the box and located at 30m from the actor is maybe not looking precisely at what the actor is doing but the use of this kind of proxy for observation is not an inherent problem. Indeed, Hoppitt (2017; S5) has shown that error in identifying observers does not increase the risk of a spurious social transmission effect but acts to make estimates of the strength of social learning conservative.

378: “have then been” ◊ “were”

379: “who are participated the most”: please specify

At some point, no new individual participated to the experiment, that is why we tested opportunistically the individuals who had been attending over the course of the experiment but who did not manipulate yet the boxes when the individuals who already learnt the task were not around. These individuals are the ones marked with a star (*) in table 1. We specified it in the text and legend (l 514-521; l 718-719).

406: why individual analyses? Why not at the group level?

We ran these binomial tests to test whether individuals had a personal preference for one of the two techniques over the course of the experiment (i.e. after having learnt the task). We added a precision (l 552-556). In KB, we do not have enough data on all group members to test the group level preference. In NH, when running a binomial test on group data, we get that NH group significantly preferred the ‘lift’ option than the ‘pull’ one ($p=0.003$; l 286).

536: “The thicker an edge is, the bigger ...”

I have two version of Figure 2, one has orange names, the other does not. Orange on green is bad for people who are color blind.

We are sorry for this mistake when uploading the file online. We changed the colors of the figure in order to be readable for color blind people.

Reviewers' Comments:

Reviewer #1:

None

Reviewer #2:

Remarks to the Author:

The authors have put a lot of effort into the revision. There is no doubt that NBDA (or the OADA) variant is a cool tool, and that the experiment is very smart. Yet, I still feel that there is room for improvement; specifically when it comes to convincing the reader that NBDA is the appropriate tool to model the acquisition of the different box opening techniques in the vervet monkeys.

Showing that one model is better than another one is one thing, but showing how accurately the model fits the data is another. There is not sufficient information to judge this. Thus, the authors need to clearly spell out the numbers of learning events they modeled. Please provide the reader with a table with the important raw data that contain for each learning event how often a given subject had seen how many other subjects had used which technique prior to learning.

It also remains unclear how the authors distinguish between social and asocial learning when an animal switches to the alternative technique: One of the key statements is that animals are 31 times faster to subsequently asocially learn the other option than individuals naïve to both options. Could the authors rule out that the individual had ever observed the alternative technique? And how can they be certain that this learning is now asocial? And did they include these events in the model for social learning? Again, I think this needs to be specified.

In this context, I was happy to see that the authors now report the latencies, but how was latency determined? Are these the latencies from first having observed a subjects using option Y until using option Y? Is this actual time or time boxes being available? The latencies are very long: what does it tell us about the learning process if the latency can be up to 24 hours? More information is needed here. It would be helpful if the latencies were also reported in hours and minutes.

In their rebuttal letter, the authors object that it would only be necessary to show [how many subjects had only seen one technique and then also used this technique] if they "were claiming that all vervets (save the innovators) learned by social transmission (copying), and/or that observation always leads to learning". I think the authors are trying to evade a critical point here. Nobody assumes that learning can only be shown when all subjects acquired the technique socially. The critical point is whether the likelihood of using one option rather than another one depends on the social input, i.e. who and how many other subjects used that particular technique before. If the majority does learn it socially, everybody will be convinced, but if it is only half of the subjects, doubts will arise. To judge this, the authors should please provide the actual numbers and not force the reader to extract these data from the raw data files.

It is impossible to judge the worthiness of the simulations because the information given is insufficient (rebuttal letter only).

Furthermore, it is important to distinguish between the opportunity to learn and the actual rate of transmission. A rate of transmission can only be inferred once it is established that the subject indeed learns from the model(s). This semantic precision is necessary to avoid conflating the premise with the result. The authors write (line 179): The underlying assumption is that the rate of social transmission between individuals is proportional to the network connection between them; the more two individuals are connected (i.e. observe each other), the more opportunities they have to learn from each other. A

parameter, s , is fitted to the data estimating the rate of transmission per unit of network connection, with representing the null hypothesis of only asocial learning.”

In summary, this is certainly a potent method, but given the small number of events, and the lack of transparency in reporting, considerable doubts remain whether the dynamic network approach is indeed helpful to model the acquisition of box opening in vervets.

Reviewers' comments:

Reviewer #2 (Remarks to the Author):

The authors have put a lot of effort into the revision. There is no doubt that NBDA (or the OADA) variant is a cool tool, and that the experiment is very smart. Yet, I still feel that there is room for improvement; specifically when it comes to convincing the reader that NBDA is the appropriate tool to model the acquisition of the different box opening techniques in the vervet monkeys.

NBDA was specifically developed for modeling the diffusion of novel behavior, so our understanding is that the reviewer is requesting more explanation of the extension of NBDA to model the diffusion of alternative techniques for opening the box. This is not a novel development - we have made this clear and explained further how the

extension to multiple options works, and how it relates to other statistical tools in the Supplemental Information (SI section 1.1), where we have added:

“A second extension of NBDA is to include multiple options for solving a task (Atton et al., 2012). In a normal OADA, the parameter values are optimised to maximise the power of the model to predict which individual will be the next to learn the target behaviour at each acquisition event. If we extend the OADA to multiple options the parameter values are optimized to predict the combination of which individual will be next to learn and which option they will learn to use (assuming the options are including in the same stratum, see below). Thus, a multi-option NBDA differs from statistical approaches that aim to infer the presence or absence of social learning based on the option used alone (Kendal et al., 2009) - rather this information is added to the pattern of spread across the network when quantifying the strength of social transmission. It is also important to note that an NBDA extended to multiple options is not intended to model the role of social learning in the development of a preference for specific options over time, once both are acquired to the repertoire (cf. experience weighted attraction models, Barrett, McElreath & Perry 2017). Rather, it is intended to model the acquisition of different behavioural variants to the behavioural repertoire, and address whether or not their acquisition to repertoire is independent (see below)”

Showing that one model is better than another one is one thing, but showing how accurately the model fits the data is another. There is not sufficient information to judge this. Thus, the authors need to clearly spell out the numbers of learning events they modeled. Please provide the reader with a table with the important raw data that contain for each learning event how often a given subject had seen how many other subjects had used which technique prior to learning.

We provide now in Table 1 the number of times each individual observed lift and pull options before their first success (while naïve) and the number of individuals observed using lift and pull options before their first success. These summarized information might show that each individual was more likely to use the option they had seen most (for example, Bela in NH observed 129 times sli and 39 spu before its first success which was sli), but the NBDA tells us we would only expect this to be the case after we have allowed for a bias towards the easier option. Other examples do not follow this expectation, as for example Malawi in KB who observed 59 times sli and 17 times spu before its first success which was spu.

However, we note that the choice of option for the vervets' first solves, and the observational experience they have accrued to that point is only part of the information used to make our inferences (our addition in the SI cited above makes it explicit what is being modeled in an OADA applied to multiple options). Therefore, we do not believe the requested information fulfills the reviewer's requirement for us to show model fit. The full data used by OADA cannot be summarized in a table of reasonable size, so we instead draw some plots representing the predicted probability of learning in function of the number of acquisition event for the best social model and the asocial model for each group (see Fig. S2 and S3 in SI). For each acquisition event, we plotted the predicted probability that each individual will be the next to learn using each option (each point is an individual x option combination). The individuals that did learn is plotted in red, and these are joined by red lines. The blue line shows the average probability across all individuals. The triangles are the individuals who have already learned the other option. You can see on the plots that the better the model fit, the more red points we would see a long way above the blue line and the fewer points we would see a long way below the blue line. We would still expect to see some points below the blue line, even with a good model fit- if there is a big cloud of points with a low probability each, it can still be likely that one of these will be the individual to learn. We can see the triangles are plotted high showing that if an individual has learned one option the model is predicting they are one of the most likely to learn the other option regardless of their observational experience. The social model has more red points higher up the plot, and fewer points well below the blue line. OADA is fitted using precisely this information: the parameter values are chosen to increase the overall height of the red points- specifically to maximize the product of their heights, which is the likelihood for the model.

It also remains unclear how the authors distinguish between social and asocial learning when an animal switches to the alternative technique: One of the key statements is that animals are 31 times faster to subsequently asocially learn the other option than individuals naïve to both options. Could the authors rule out that the individual had ever observed the alternative technique? And how can they be certain that this learning is now asocial? And did they include these events in the model for social learning? Again, I think this needs to be specified.

We do not a priori identify which events are asocial learning and which are social transmission. The model assumes that any learning event could have occurred by asocial learning or social transmission and models the relative rate at which each individual learns each option by each process, as a function of the predictor variables, as shown in our model definition. The model finds strong evidence of an effect of having previously solved the other option on the rate of asocial learning, this corresponds to a 31 x increase in the rate of solving by asocial learning. Therefore, in answer to the specific questions posed: i) we do not rule out the possibility that individuals

had observed this technique, since the potential effects of such observations are included in the model; ii) we cannot be certain that these events are occurring by asocial learning, but this is not necessary to make the inference, iii) yes all events were included in the OADA model. What the model is picking up is that if an individual has learned one option it is likely that it will learn the other option soon *regardless* of its observational experience.

In other words, we report an effect of having solved the other option on the rate of ASOCIAL learning - because this is the "in words" interpretation of what is happening in our fitted model. However, we appreciate that we have not considered the possibility that the underlying statistical effect might be the result of an increase in social learning rather than in asocial learning, rendering our conclusion premature. To address this, we replaced the effect of otherOption on asocial learning with an effect on social learning, and found an increase of 6.63 units in AICc, indicating the effect is much better explained by an increase in asocial learning rate. This must be because the learning rate for learning a second option is increased regardless of observational experience, as opposed to being increased in proportion to the number of observations of the second option. We also considered models in which there was an effect of otherOption on both asocial learning and social learning, both constrained to be the same (increase of 2.17 units AICc) and allowed to be different (increase of 1.79 units AICc), indicating no evidence that the effect extended to social learning as well as asocial learning. We report these additional analyses in section 1.5 of the SI and refer to this on line 223 of the main text.

In this context, I was happy to see that the authors now report the latencies, but how was latency determined? Are these the latencies from first having observed a subjects using option Y until using option Y? Is this actual time or time boxes being available? The latencies are very long: what does it tell us about the learning process if the latency can be up to 24 hours? More information is needed here. It would be helpful if the latencies were also reported in hours and minutes.

The latencies that we provided in the previous corrected version of the manuscript were the latencies of the first contact to one of the boxes for every individual. For this, we recorded at which time the boxes were put on the ground at the beginning of each experiment and we recorded for each individual, the exact time at which they contacted the boxes. We also recorded the exact duration of every experiment. As an example, the first contact latency for an individual who contacted one box for the first time during experiment 3 is the cumulative time of duration of experiment 1 (which corresponds to the duration when boxes were provided to the group) + duration of experiment 2 + latency of its first contact in experiment 3. This is the reason why some latencies are very long, because some individuals contacted one box for the first time several weeks after the first experiment. We reported now in the present revised version the latencies of the first success for every individual as well. These latencies of first success were calculated following the same calculation as for the first contact. As requested, we reported the latencies also in hours: minutes: seconds (Table 1; l 531-538).

The fact that the latencies are long could be seen as being consistent with social transmission - if vervets go a long time without solving the task, until one individual happens to solve it, which triggers the diffusion by social transmission. The other variant of NBDA, time of acquisition diffusion analysis (TADA) would pick up on such patterns. However, this pattern could also be explained by other factors, such as a decrease in neophobia over time causing an acceleration in asocial solving rate over time. Alternatively, it could be that the group is synchronized in their solving rate due to external factors - if the weather conditions/ lack of predators on a particular day caused many individuals to start solving the task. Both of these effects, and others, could result in a spurious social transmission effect (see Hoppitt, Boogert and Laland 2013). Consequently, we prefer to use an OADA which ignores the time course of the events and is sensitive only to the order of events as explained in the SI section 1.1. This is also the technique most commonly used to analyze time to event data in general (OADA is an extension of the Cox survival analysis model).

In their rebuttal letter, the authors object that it would only be necessary to show [how many subjects had only seen one technique and then also used this technique] if they "were claiming that all vervets (save the innovators) learned by social transmission (copying), and/or that observation always leads to learning". I think the authors are trying to evade a critical point here. Nobody assumes that learning can only be shown when all subjects acquired the technique socially. The critical point is whether the likelihood of using one option rather than another one depends on the social input, i.e. who and how many other subjects used that particular technique before. If the majority does learn it socially, everybody will be convinced, but if it is only half of the subjects, doubts will arise. To judge this, the authors should please provide the actual numbers and not force the reader to extract these data from the raw data files.

We had not intended to evade the point, we misunderstood what the reviewer was requesting. We had understood the reviewer was requesting we conduct another analysis to see whether the likelihood of using one option was dependent on social input. We responded by saying that this information is already taken into account

in the multi-option OADA we conduct. To clarify further, it is unlikely that an analysis of option choice alone would yield a conclusive result, but when combined with the data on the order of events we are able to take advantage of all the data available to us. We have explained more clearly how the multi-option OADA combines this information in section 1.1 of the SI (see above).

It is clear now that the reviewer was instead requesting that we provide the numbers to show how first option choice related to observational experience, which we now do in Table 1. However, as we explain above, we do not feel this reflects the fit of the OADA models directly, since this is only part of the information used to make inferences - so we have also provided figures S2-S3 (see above).

It is impossible to judge the worthiness of the simulations because the information given is insufficient (rebuttal letter only).

Information and results concerning the simulations requested by the reviewer are provided in the Supplemental information file (1.2. Validation of method detecting option-specific social transmission; p7-8) and in the annotated script used to run these simulations (Canteloup_et_al_OptionBias simulations.R).

Furthermore, it is important to distinguish between the opportunity to learn and the actual rate of transmission. A rate of transmission can only be inferred once it is established that the subject indeed learns from the model(s). This semantic precision is necessary to avoid conflating the premise with the result. The authors write (line 179): The underlying assumption is that the rate of social transmission between individuals is proportional to the network connection between them; the more two individuals are connected (i.e. observe each other), the more opportunities they have to learn from each other. A parameter, s , is fitted to the data estimating the rate of transmission per unit of network connection, with representing the null hypothesis of only asocial learning.”

In an NBDA, the network connection quantifies the opportunities for social transmission (this is the model assumption). The rate of transmission is ($s * \text{network connection}$) so is not the same as the opportunities for social learning. If we understand correctly, the reviewer is concerned that the stated model assumption means that we are assuming, a priori, that social transmission is taking place. This is not the case, because $s=0$ represents the case where no social transmission is occurring: if the 95% CI for s includes zero, we conclude that there is not strong evidence against the hypothesis of no social transmission.

Therefore, we are no more conflating the premise with the result that when one fits a linear regression. Here the assumption is that X and Y are linearly related, i.e. $Y = \beta * X + \text{error}$. This does not a priori assume the result that Y is related to X , because $\beta=0$ represents the case where they are unrelated. If the 95% CI for β includes 0, one concludes there is not strong evidence for a relationship between Y and X .

To avoid this confusion, we have amended the section of concern to state “The underlying assumption is that the rate of social transmission between individuals, **if it occurs**, is proportional to the network connection between them” (l 179) such that it is clear at this point that we are not a priori assuming that social transmission must be occurring.

In summary, this is certainly a potent method, but given the small number of events, and the lack of transparency in reporting, considerable doubts remain whether the dynamic network approach is indeed helpful to model the acquisition of box opening in vervets.

Reviewers' Comments:

Reviewer #2:

Remarks to the Author:

Thanks for the authors for trying to clarify what they did. Figures S2 and S3 helped me to identify where the misunderstanding between the authors and me may have been rooted. Inspecting Fig. S2 and S3 indicate that the model predicts the likelihood for each subject that it will learn, that is, the sample size is correspondingly larger. In other words, the model appears to be predictive in the sense that it assigns learning probabilities to different subjects, updating after each event the learning probability. This is very different from an approach that starts with the learning event and then retrospectively reconstructs what the input was and whether this is in line with CO or OS learning (which is what I had assumed, resulting in an N of 14). If I am correct with this diagnosis, the authors might want to clarify this in the manuscript.

The predicted values shown in these supplementary figures show that the probability of learning is substantially higher when the animals already know one technique; is this the basis for the conclusion that these animals learn the new technique individually 31 times faster? If this is the case, the authors may want to write that the likelihood to learn is 31 times higher. 'Faster' may be misleading here, as the model is not based on actual temporal measurements. The details about who touched what when can then be moved to the SI, as it does not really appear to be relevant.

Figures S2 and S3: the red line needs to go; lines between points should only be used for repeated measurements.

I strongly recommend to present Figures S2 and S3 (at least for the best social model) in the main part of the manuscript.

Because of the size limit of the paper and because we added one figure with some explanation, we moved this part to the SI.

Figures S2 and S3: the red line needs to go; lines between points should only be used for repeated measurements.

We removed the red lines in the plots.

I strongly recommend to present Figures S2 and S3 (at least for the best social model) in the main part of the manuscript.

We added one figure (Fig. 3) depicting the plots for the best social models in the manuscript.